# Causal Effects of 25-Hydroxyvitamin D on Metabolic Syndrome and Metabolic Risk Traits: A Bidirectional Two-Sample Mendelian Randomization Study

**DOI:** 10.3390/biomedicines13030723

**Published:** 2025-03-15

**Authors:** Young Lee, Je Hyun Seo, Junyong Lee, Hwa Sun Kim

**Affiliations:** 1Veterans Medical Research Institute, Veterans Health Service Medical Center, Seoul 05368, Republic of Korea; lyou7688@gmail.com (Y.L.); jazmin2@naver.com (J.H.S.); 2Department of Family Medicine, Veterans Health Service Medical Center, Seoul 05368, Republic of Korea; talker69@daum.net

**Keywords:** metabolic syndrome, 25-hydroxyvitamin D, multivariable mendelian randomization, inverse–causal relationship

## Abstract

**Background/Objectives:** Individuals with metabolic syndrome (MetS) present reduced 25(OH)D levels. We performed a two-sample Mendelian randomization (MR) study to investigate whether causal relationships exist between 25(OH)D levels and MetS/MetS risk traits, including waist circumference, body mass index (BMI), hypertension (systolic/diastolic blood pressure), triglyceride, high-density lipoprotein cholesterol, and glucose levels. **Methods:** We employed genetic variants related to 25(OH)D levels from the SUNLIGHT Consortium and a European genome-wide association study meta-analysis, including UK Biobank (UKB) data, as well as variants for MetS and MetS risk traits from UKB and multiple European consortia. Several MR methods were used, i.e., inverse-variance weighted, weighted median, and MR–Egger regression. Heterogeneity and horizontal pleiotropy analyses were performed to ensure the stability of candidate single-nucleotide polymorphisms (SNPs) as the instrumental variable. We first conducted univariable MR to investigate the relationship between 25(OH)D levels and MetS, including its related risk traits, and subsequently performed multivariable MR to adjust for potential confounders. **Results:** This study did not provide evidence of a causal relationship between 25(OH)D levels and MetS/MetS risk traits. However, we found that several risk traits of MetS, such as waist circumference, BMI, and TG, had an inverse–causal relationship with 25(OH)D levels, suggesting that 25(OH)D levels could be secondary consequences of metabolic illnesses. **Conclusions:** We identified no causal relationship between 25(OH)D levels and MetS/MetS risk factors. However, 25(OH)D levels may result from MetS traits.

## 1. Introduction

Metabolic syndrome (MetS) has become one of the most common health problems worldwide, and it is related to age, sex, and ethnicity [1,2,3,4,5]. Although the cutoff for MetS criteria differs among the guidelines of the World Health Organization (WHO), National Cholesterol Education Program-Adult Treatment Panel III (NCEP-ATP III), and International Diabetes Federation (IDF), it includes high blood pressure, hyperglycemia, low levels of high-density lipoprotein (HDL) cholesterol, hypertriglyceridemia, and abdominal obesity causing insulin resistance [6,7,8,9,10]. Individuals with MetS have a higher risk of developing type-2 diabetes mellitus, cardiovascular disease, and stroke, which can lead to disability and/or early death [11,12,13]. Moreover, the increasing prevalence of obesity and associated metabolic disorders, including MetS, has prompted urgent investigations into the underlying factors underlying their progression [14,15,16,17].

The major biological role of 25(OH)D is to aid in mineral homeostasis, including that of calcium and phosphorus, ultimately facilitating proper bone formation [18]. In addition to its classic role in bone formation, numerous studies have highlighted its non-classic role in certain diseases such as infection, neurodegenerative diseases, and metabolic illnesses [19,20,21,22]. Low serum 25(OH)D levels are associated with various metabolic illnesses, such as non-alcoholic fatty liver disease, obesity-related cancer, insulin resistance, and cardiovascular disease [22,23,24,25,26]. Notably, initial efforts underscore a significant role of 25(OH)D, not only as an essential nutrient but also as a potential therapeutic target and modifiable risk factor in the management of these metabolic illnesses, including MetS. Investigating the causal relationship between 25(OH)D and these conditions is crucial. However, further evidence is still needed to clarify whether 25(OH)D levels have a relationship with MetS and its associated risk traits [27,28].

Clinicians and researchers have designed methods for clarifying the causal relationships between exposure and health outcomes to overcome biases in observational studies that are limited by the interference of multiple known and unknown confounders [29,30,31]. Mendelian randomization (MR) was introduced to address the limitations of observational studies and is increasingly used to explore causality based on observational results [32,33]. MR analysis utilizes single-nucleotide polymorphisms (SNPs), genetic variants robustly associated with exposure in genome-wide association studies (GWAS), as instrumental variables (IVs) to understand the influence of exposure on various health, social, and economic outcomes [30,34,35,36]. Because each individual’s genetic variants are randomly inherited and predetermined at birth, they cannot be influenced by different confounders related to the association between exposure and outcomes [37,38]. Thus, MR studies have the potential to eliminate the inherent biases and confounders present in traditional observational studies.

This study aims to bridge the existent gap in knowledge, namely, understanding the causal relationship between 25(OH)D and MetS, along with its risk traits. Although many observational studies have found correlations, there is insufficient evidence to firmly establish causality. This gap presents a critical limitation in fully understanding the implications of 25(OH)D in the context of MetS. Here, we address this gap by conducting a two-sample MR analysis using the latest large-scale datasets to explore the causal relationship between 25(OH)D and MetS, or its risk traits, thereby contributing valuable insights to future health intervention strategies.

## 2. Materials and Methods

### 2.1. Data Resources

The datasets for the GWAS summary statistics used in this analysis were obtained from the GWAS Catalog, Pan-UK Biobank, FinnGen, and other large-scale consortia. A schematic of the study flow is shown in Figure 1. For the exposure and outcome datasets, we employed SNPs associated with 25(OH) D levels from the SUNLIGHT Consortium and those related to MetS or each risk trait of MetS (waist circumference (WC), triglyceride (TG), HDL cholesterol, systolic/diastolic blood pressure (SBP/DBP), and glucose levels) from the UK Biobank [39,40]. Both forward and reverse MR analyses were initially performed using this dataset (*dataset 1*). However, owing to the low genetic contribution to the variation in 25(OH)D levels reported by the SUNLIGHT Consortium, this dataset was utilized solely for reverse MR analysis to support our findings. To complement this, the second exposure and outcome dataset (*dataset 2*) was added, leveraging SNPs associated with 25(OH)D levels from meta-analyses of UKB and other European GWAS data. This second dataset allowed us to perform both forward and reverse MR analyses. The outcome traits in this dataset included MetS and its associated risk factors, such as WC, body mass index (BMI), TG, HDL cholesterol, hypertension, and glucose levels, obtained from various large-scale consortia. Additionally, confounder data were incorporated to account for potential biases in both forward and reverse MR analyses. For reverse MR analysis, confounders such as skin color and physical activity (walking) were obtained from the UKB. For forward MR analysis, confounders, including physical activity (strenuous sports), alcohol consumption, and smoking, were also sourced from the UKB. The summary statistics of the data resources are listed in Table 1.

### 2.2. Selection of SNPs as IVs

In MR studies, intense IVs should be used to ensure a significant correlation between exposure and outcomes and minimize any possible weak IV bias [47]. We selected SNPs as IVs satisfying the following conditions: (i) SNPs associated with exposure at the genome-wide significance threshold (*p* < 5 × 10^−8^) were used as IVs; (ii) SNPs were pruned for linkage disequilibrium (LD; *r*^2^ < 0.001; clumping distance, 10,000 kb) to ensure independence among the IVs, and LD calculations utilized the 1000 Genomes Phase III European dataset as a reference; (iii) to verify the reliability of each genetic instrument, we assessed their F-statistics for univariable MR as follows:F = R^2^(n − 2)/(1 − R^2^)(1)
where *R*^2^ is the proportion of exposure variance by genetic variance, and *n* is the sample size [48]. An F-value of >10 suggests that the causal estimates are likely not biased by weak instruments [49]. IV strength was assessed for the multivariable MR (MVMR) analysis using conditional F-statistics; values greater than 10 similarly indicated adequate instrument strength for the analysis [50].

### 2.3. MR Study

TwoSampleMR, simex, and MVMR packages (version 3.6.3) in R (version 3.6.3; R Foundation for Statistical Computing, Vienna, Austria) were used to analyze the data of the MR study. MR results were interpreted based on the following assumptions: (i) genetic variants should be significantly correlated with exposure (in multivariable analysis, it is necessary but not sufficient for each exposure to be strongly predicted by IVs, given the other exposures included in the model); (ii) these variants should be unrelated to any confounder of the exposure–outcome relationship; and (iii) these variants should solely affect the outcomes through exposure, showing no directional horizontal pleiotropy effect. Heterogeneity may also indicate the pleiotropic effects of genetic variations [51]. Thus, it is critical to evaluate heterogeneity (*p* < 0.05, significant heterogeneity of individual SNP) or pleiotropy (*p* < 0.05, strong possibility of pleiotropy of each SNP) of SNPs to implement the MR study. The Cochran’s Q test from inverse-variance weighted (IVW) and Rücker’s Q′ test from MR–Egger were employed to test the heterogeneity of candidate SNPs, and a no-measurement-error (NOME, I^2^) test was conducted to stabilize the validity of IVs. The IVW method was used as the primary analysis, and the weighted median and MR–Egger regression, with or without adjustment via the Stimulation Extrapolation (SIMEX) approach, and MR-PRESSO were applied as supplementary methods for the MR analysis [52]. The IVW analysis is most plausible when all genetic variations satisfy the three assumptions for IVs [30]. The weighted median method provides plausible results even when some IVs are invalid (<50%) [37]. The MR–Egger method can obtain preferable causal effects even when pleiotropic effects exist by correcting for horizontal pleiotropy [53]. If the NOME assumption is violated (I^2^ < 90%), then MR–Egger (SIMEX) is suitable for addressing the bias. The MR-PRESSO test has an advantage over MR–Egger when horizontal pleiotropy is present in that it identifies and removes pleiotropic SNPs [54]. Consequently, the findings were interpreted based on suitable univariable MR approaches [53,55,56]. For the MVMR analysis, which allows for the simultaneous evaluation of multiple exposures, we used MVMR IVW to adjust for confounders and isolate the relation between 25(OH) D levels and MetS, including its risk traits [50]. Heterogeneity and potential pleiotropy among IVs were assessed using the Q_A_ statistic, a refinement of Cochran’s Q [50]. In cases where the conditional F or Q_A_ statistics suggested weak instruments or possible pleiotropy, a Q-minimization method (Q-het) was employed to derive robust causal estimates, complementing the MVMR-IVW framework. Standard errors were computed using the jackknife method [50]. For exposures with overlapping samples, calculating the conditional F and Q_A_ statistics required covariance estimates for the effect of each SNP on each exposure. These covariances were obtained via a phenotypic correlation matrix derived from the intercept of bivariate LD score regression [57,58,59]. Statistical significance was set at *p* < 0.05.

## 3. Results

### 3.1. Genetic IVs in Univariable MR

To clarify the causal relationship between 25(OH)D levels and MetS, including its risk traits, we conducted a two-sample MR analysis. Several statistical approaches were applied to select SNPs as IVs. A schematic representation of the study flow for the selection of IVs is presented in Figure 1. In *dataset 1* (reverse direction), 71 SNPs were selected as IVs for MetS. Additionally, 318, 277, 327, 244, 229, and 114 SNPs were extracted as IVs for MetS risk traits, namely WC, TG, HDL cholesterol, SBP, DBP, and glucose, respectively. *For dataset 2* (forward direction), a total of 89, 92, 93, 104, 104, 101, and 104 SNPs were selected as IVs for 25(OH)D levels when MetS, WC, BMI, TG, HDL cholesterol, hypertension, and glucose were the outcomes, respectively. In *dataset 2* (reverse direction), 543, 41, 67, 369, 414, 268, and 71 SNPs were selected as IVs for MetS, WC, BMI, TG, HDL cholesterol, hypertension, and glucose, respectively. Each set of IVs demonstrated significant genome-wide associations (*p* < 5 × 10^−8^) with exposure traits and was not in the LD within 10,000 kb. The F values for all selected SNPs exceeded 10, indicating strong IVs with a low probability of weak instrument bias (Table 2 and Appendix A).

### 3.2. Heterogeneity and Pleiotropy Tests for IVs in Univariable MR

We conducted Cochran’s Q test from IVW and Rücker’s Q’ test from MR–Egger for heterogeneity, as well as horizontal pleiotropy analyses, to assess the stability of candidate SNPs as IVs.

In forward MR analyses, the assumption of NOME was satisfied (I^2^ > 90) (Table 2, *dataset 2*). Significant heterogeneity was observed across all outcomes (Cochran’s Q, all *p* < 0.001; Rücker’s Q’, all *p* < 0.001). Despite this heterogeneity, the MR–Egger regression intercepts indicated no horizontal pleiotropy for most outcomes, with non-significant *p*-values for waist circumference (*p* = 0.199), BMI (*p* = 0.208), HDL (*p* = 0.650), hypertension (*p* = 0.703), and glucose (*p* = 0.805). However, a pleiotropic effect was detected for MetS (MR–Egger intercept *p* = 0.012; SIMEX-adjusted *p* = 0.013) and TG (MR–Egger intercept *p* = 0.028; SIMEX-adjusted *p* = 0.030). The MR-PRESSO global test was significant across all outcomes. Consequently, it is recommended to use MR-PRESSO for all outcomes [39].

In reverse MR analyses, the assumption of NOME was satisfied (I^2^ > 90) except when MetS, WC, and DBP were exposed (Table 2). In *dataset 1*, no heterogeneity was observed for WC (Cochran’s Q, *p* = 0.190; Rücker’s Q’, *p* = 0.198) or SBP (Cochran’s Q, *p* = 0.070; Rücker’s Q’, *p* = 0.079), and no pleiotropic effects were detected in the MR-PRESSO global test (WC, *p* = 0.185; SBP, *p* = 0.072). Conversely, when MetS, TG, HDL cholesterol, DBP, and glucose levels were exposure factors, heterogeneity of IVs was apparent (Cochrane’s Q, all *p* < 0.05; Rücker’s Q′, all *p* < 0.05), and the MR-PRESSO global test was significant. Despite this heterogeneity, the MR–Egger regression intercepts also revealed no horizontal pleiotropic effects (all *p* > 0.05) regardless of SIMEX adjustment (Table 2, *dataset 1*). Consequently, the IVW approach was used to determine the causality between WC or SBP and 25(OH)D levels. The effects of MetS and DBP on the 25(OH)D levels were evaluated using the MR–Egger (SIMEX) approach. The MR-PRESSO method was used to investigate the effects of TG, HDL cholesterol, and glucose levels on 25(OH)D levels [51]. For TG exposure to 25(OH)D, an MR-PRESSO outlier test was conducted; however, no outliers were found. In *dataset 2*, significant heterogeneity was observed across all outcomes (Cochran’s Q, all *p* < 0.001; Rücker’s Q’, all *p* < 0.001). Despite this heterogeneity, the MR–Egger regression intercepts indicated no horizontal pleiotropy for most outcomes except glucose. The MR-PRESSO global test was significant across all outcomes. Consequently, it is recommended to use MR-PRESSO for most outcomes, except MetS and WC. The effects of MetS and WC on 25(OH)D levels were evaluated using the MR–Egger (SIMEX) approach because I^2^ < 90 [51].

### 3.3. Effect of 25(OH)D on Metabolic Syndrome and Its Risk Factors in Univariable MR

Forward MR analyses (*dataset 2*) using MR-PRESSO as the main method revealed significant causal associations between 25(OH)D and specific metabolic outcomes, particularly MetS and TG (Figure 2). For MetS, MR-PRESSO identified a significant negative association (odds ratio [OR] = 0.96; 95% confidence interval [CI] 0.93–0.98; *p* = 0.004), consistent with the IVW method (OR = 0.93; 95% CI 0.89–0.99; *p* = 0.014), indicating that higher 25(OH)D levels may reduce the risk of MetS. Similarly, for TG, MR-PRESSO demonstrated a significant negative effect (β = −0.19; 95% CI −0.24 to −0.14; *p* < 0.001), aligned with IVW (β = −0.27; 95% CI −0.42 to −0.11; *p* < 0.001), suggesting a potential role of 25(OH)D in lowering TG levels. In contrast, no significant associations were observed for WC (β = 0.06; 95% CI −0.01–0.12; *p* = 0.099), BMI (β = 0.0005; 95% CI −0.06–0.06; *p* = 0.986), HDL (β = −0.04; 95% CI −0.09–0.0007; *p* = 0.059), hypertension (OR = 0.96; 95% CI 0.89–1.03; *p* = 0.258), or glucose (β = −0.001; 95% CI −0.03–0.02; *p* = 0.938). The scatter plots of the forward MR analysis are shown in Figure 3.

### 3.4. Effect of Metabolic Syndrome on 25(OH)D Levels on Univariable MR

In the reverse MR analyses for *dataset 1*, the MR–Egger (SIMEX) method did not reveal a significant causal association between MetS and 25(OH)D (β = 0.01; 95% CI −0.01–0.04; *p* = 0.336), as shown in Figure 4. Additionally, no significant causal associations were identified between MetS and 25(OH)D levels using the IVW, weighted median, and MR–Egger methods, with *p*-values of 0.819, 0.800, and 0.325, respectively. However, WC, one of the risk traits of MetS, demonstrated significant inverse causality on 25(OH)D levels using IVW (β = −0.03; 95% CI −0.05 to −0.01; *p* = 0.001), although other MR approaches did not yield significant results. The IVW method was primarily employed because the IVs for WC satisfied the assumptions. Additionally, suitable primary MR approaches were conducted for the causality of other MetS risk traits on 25(OH)D levels using assumptions for IVs [51]. The IVW approach for SBP (β = 0.01; 95% CI −0.003–0.03; *p* = 0.111), MR–Egger (SIMEX) approach for DBP (β = 0.02; 95% CI −0.04–0.08; *p* = 0.564), and MR-PRESSO method for genetic variants of HDL cholesterol (β = 0.01; 95% CI; −0.002–0.02; *p* = 0.136) and glucose (β = −0.0008; 95% CI −0.02–0.02; *p* = 0.925) were conducted to assess the effect of metabolic risk traits on 25(OH)D. For TG genetic variants, the MR-PRESSO approach was recommended according to assumptions of IVs but identified no outliers; the TG traits showed no causal effect on 25(OH)D levels across any MR method (all *p* > 0.05, Figure 4).

By contrast, the results for *dataset 2* revealed significant associations between several exposures and 25(OH)D levels (Figure 5). For MetS, the MR–Egger (SIMEX) method indicated a significant negative association (β = −0.11; 95% CI −0.19 to −0.03; *p* = 0.005), consistent with other methods. For WC, MR–Egger (SIMEX) yielded non-significant results (β = −0.01; 95% CI −0.14–0.11; *p* = 0.847), whereas the IVW, weighted median, and MR-PRESSO methods showed significant negative associations. For BMI, glucose, and TG, all methods, including MR-PRESSO, consistently showed significant negative associations with 25(OH)D (BMI: β = −0.06; 95% CI −0.09 to −0.04, *p* < 0.001; glucose: β = −0.08; 95% CI −0.12 to −0.05, *p* < 0.001; TG: β = −0.12; 95% CI −0.14 to −0.10, *p* < 0.001). For HDL, the weighted median method showed a significant negative association (β = −0.03; 95% CI −0.05 to −0.01; *p* = 0.012), whereas other methods, including MR-PRESSO, yielded non-significant results. For hypertension, no significant associations were observed across all methods.

A scatter plot illustrating the genetic associations between MetS and its risk traits against genetic associations with 25(OH)D for *dataset 1* is shown in Figure 6, whereas Figure 7 presents the corresponding results for *dataset 2*.

### 3.5. Multivariable MR

The MVMR analyses revealed no significant associations in the forward direction but demonstrated significant associations in the reverse direction, with Model 2 results being considered more reliable due to fewer but more relevant confounder adjustments (Table 3). Specifically, in forward direction analyses (*dataset 2*), Model 1 included adjustments for physical activity (strenuous sports), alcohol consumption, and smoking, whereas Model 2 only accounted for smoking. Physical activity (strenuous sports) and alcohol consumption were excluded in Model 2 owing to their conditional F-statistics being <5. Similarly, in reverse direction analyses, Model 1 incorporated adjustments for physical activity (walking) and skin color, whereas Model 2 included only skin color. Although the methods used in this study are robust for weak instruments, they cannot produce reliable estimates when the instruments become very weak [50]; therefore, Model 2 was considered more appropriate for interpretation. In *dataset 1* (reverse direction), only WC showed a significant negative association with 25(OH)D in Model 2 (β = −0.036; 95% CI −0.065 to −0.006), whereas other metabolic factors, including TG, HDL, SBP, DBP, and glucose, did not exhibit significant associations. By contrast, *dataset 2* (reverse direction) demonstrated significant negative associations for MetS (Model 2: β = −0.141; 95% CI −0.175 to −0.107), WC (β = −0.097; 95% CI −0.143 to −0.051), BMI (β = −0.059; 95% CI −0.095 to −0.024), and TG (β = −0.120; 95% CI −0.164 to −0.077), supporting a causal effect of these metabolic factors on lowering 25(OH)D levels, whereas HDL, hypertension, and glucose showed no significant associations. Conditional F-statistics for exposures in each model and the heterogeneity statistics are presented in Appendix A.

## 4. Discussion

In this study, we explored the causal relationship between 25(OH)D levels and either the risk of MetS or each MetS risk trait using instrumental SNPs related to 25(OH)D levels or MetS and its risk traits (BMI, WC, hypertension (SBP/DBP), TG, HDL cholesterol, and glucose levels). Numerous studies have addressed a plausible relationship between low 25(OH)D levels and metabolic disorders, including MetS, in various populations. Vitamin D deficiency appears to be associated with hypertriglyceridemia and low HDL cholesterol levels in postmenopausal women, which increases the risk of MetS [60]. A study of individuals aged >65 years with low 25(OH)D levels showed a higher risk of MetS [61]. Huang et al. reported that nondiabetic young adults with vitamin D deficiency have a high risk of MetS, including hypertriglyceridemia, low HDL cholesterol levels, and high LDL cholesterol concentrations [62]. Similarly, Zhu et al. reported that a study population aged 17–70 years showed a linear relationship between 25(OH)D levels and serum glucose and lipid levels, concluding that higher 25(OH)D levels were related to better metabolic traits in urban Shanghai residents in China [63].

Although various studies have shown an association between 25(OH)D levels and MetS and its risk traits, our MR study could not confirm their causal relationship. Instead, it provided evidence of inverse causality with MetS and several of its components, such as BMI, WC, and TG, on 25(OH)D levels. These findings are supported by reports from several other studies. The levels of 25(OH)D vary depending on age, sex, season, diet, residential area, clothing, sunscreen use, and habits (exercise, tobacco, and alcohol), causing multiple confounders in epidemiologic studies [64,65,66,67]. Mehri et al. pointed out that the absence of long-term follow-up data could not define a causal relationship between exposure and outcomes in observational studies, thus requiring further research beyond observational studies [68]. Therefore, some studies have explored the causality between 25(OH)D levels and MetS or its metabolic risk traits. Skaaby et al. reported no statistically significant causal relationship between 25(OH)D levels and MetS in a study of specific genetic variants in their study population [69]. Similarly, Chen et al. found no evidence that a genetically determined reduction in 25(OH)D levels may increase the risk of MetS or its metabolic traits in a community-dwelling population of 10,655 individuals [70], thus supporting our findings. However, we additionally found that several components of MetS had an inverse and significant causal relationship with 25(OH)D, suggesting that the level of 25(OH)D is a secondary consequence of metabolic disorders such as obesity, non-alcoholic fatty liver disease, and insulin resistance [71,72,73,74].

Genetic variants of body mass index (kg/m^2^) were negatively associated with 25(OH)D variants in a multiple-cohort MR study [75]. Additionally, some studies have reported that homeostatic model assessment for insulin resistance was negatively correlated with 25(OH)D in adults with obesity and that impaired hepatic 25-hydroxylation in patients with non-alcoholic fatty liver disease resulted in lower levels of 25(OH)D, which also supports our findings. Xiao et al. showed the inverse causality of 25(OH)D on DBP, one of the risk factors for MetS, based on their one-sample MR analysis to assess the causal role of 25(OH)D on MetS [74]. The underlying mechanisms of 25(OH)D and MetS or its risk traits are very complex, even in a particular population, given the heterogeneity in demographic characteristics and genetic diversity in traits. It is also important to consider the biological mechanisms that are associated with MetS components, such as obesity, insulin resistance, and inflammation, that contribute to lower 25(OH)D levels to strengthen our findings. Obesity has been known to lead to 25(OH)D sequestration in adipose tissue to lower circulating levels of 25(OH)D [76]. Additionally, Manoppo et al. reported that the interplay among 25(OH)D, lipid metabolism, and inflammation may create a loop wherein metabolic dysfunction worsens 25(OH)D deficiency, suggesting that the relationship between 25(OH)D and MetS is more complex than a linear one [26]. This complexity may explain the different results reported across studies [65,67]. Therefore, further research, such as RCTs, is required to fully understand the role of 25(OH)D in metabolic traits, including reverse causality. MR analysis, such as that used in the present study, has potential strengths, including in RCT methods, for assessing the causality between exposures and outcomes in cohort studies [32,38]. Large-scale prospective studies based on larger GWAS datasets are required to expand the IVs to determine their causal relationships.

The main advantage of this study is that it reinforces the presence of inverse causality of metabolic traits on 25(OH)D levels, providing some plausible evidence of MVMR results from two different datasets (*dataset 1* and *dataset 2* for reverse direction). Additionally, to reduce the impact of population stratification, we selected large-scale genomic datasets from individuals of European ancestry in a two-sample MR study. Furthermore, our study aimed to account for confounding factors by employing MVMR, which provides a framework to adjust for some pleiotropic pathways and confounders, thus contributing to a more reliable causal interference.

One potential limitation was the presence of a certain overlap in dataset sources (e.g., UKB) between confounders and exposure or outcome datasets, which caused bias in our analysis. However, we believe that this does not pose a significant risk of challenging the presence of reverse causality on 25(OH)D. To strengthen the genetic instruments, we implemented conditional F statistics for each instrument in our MVMR approach, reducing the risk of bias from weak IV issues [77]. Additionally, the individual risk traits comprising MetS and 25(OH)D were independent in our analysis, further minimizing the risk of weak IV bias. Another limitation was that our MVMR analysis did not incorporate a comprehensive range of confounders, particularly those related to environmental and socioeconomic factors, owing to the use of summary-level data rather than individual-level data. While the method accounted for some confounding through genetic instruments, the limited scope of confounders could influence the robustness of causal interferences. Additionally, the heritability and genetic variability in 25(OH)D levels remain important considerations. Twin and family studies have reported the heritability of 25(OH)D levels to be 43–80% [78]. Despite this, *dataset 1*, which exhibited a low variance of genetic instruments for 25(OH)D from the SUNLIGHT Consortium, was insufficient to demonstrate robust MR results for the causality on MetS. Hiraki et al. suggested that multiple genes for 25(OH)D with small effects contribute to its variability [79]. Additionally, Manousaki et al. reported that some socioeconomic traits partially share heritability with 25(OH)D levels [41]. Thus, when genetic variants for 25(OH)D levels are used as the exposure, even with large-scale GWAS for 25(OH)D levels, these potential risks of confounding and pleiotropy might not be totally excluded to derive a forward causality of 25(OH)D levels in metabolic traits. In this large-scale MR analysis, our results are restricted to a European population in order to minimize bias from a diverse, mixed population, which may limit the generalizability of our findings to other ethnic populations. Genetic variations in 25(OH)D metabolism, dietary patterns, and differences in sun exposure can significantly influence 25(OH)D status across populations [80,81,82,83,84]. For instance, in the northern Chinese population, 25(OH)D levels were inversely related to the metabolic risk profiles, while in Asian Indians, who have a high prevalence of 25(OH)D deficiency, no significant association with metabolic syndrome or insulin resistance was observed. Studies in East Africans address the complex interaction between 25(OH)D levels and metabolic health [85,86,87]. Future studies should aim to explore the relationship between 25(OH)D and MetS in more diverse populations. Some studies suggest that clinical intervention with 25(OH)D may improve insulin sensitivity and reduce inflammation in individuals with obesity or type 2 diabetes [85,88]. This also emphasizes the need for further research to assess potential benefits in specific groups.

MR studies have the advantage of elucidating causal relationships between exposure and outcome. However, SNPs as genetic variants account for only a portion of the overall variance in exposure. The GWAS SNP data for 25(OH)D, MetS, and metabolic risk traits that we employed were not representative. Therefore, further studies involving a broader range of populations are needed to strengthen the present study.

## 5. Conclusions

This study used two-sample MR analysis to explore the causality between 25(OH)D and MetS. We found that the 25(OH)D level is a secondary consequence of metabolic traits rather than a causal factor. Large-scale prospective studies based on larger GWAS datasets are required to expand instrumental variables and assess causal relationships, with a focus on integrating more comprehensive datasets that include both genetic and environmental factors, improving the validity of genetic instruments.

## Figures and Tables

**Figure 1 biomedicines-13-00723-f001:**
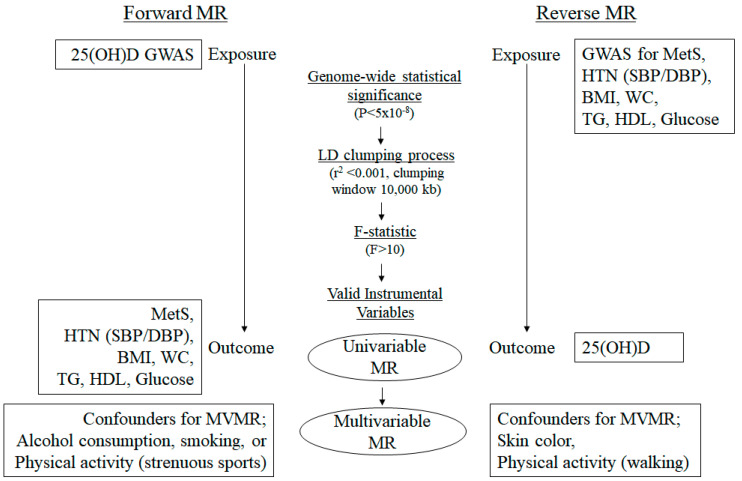
Overview of instrumental variable selection and the workflow of two-sample MR analyses. 25(OH)D, 25-hydroxyvitamin D; MetS, metabolic syndrome; HTN; hypertension, SBP; systolic blood pressure; DBP, diastolic blood pressure; WC, waist circumference; TG, triglyceride; HDL, high-density lipoprotein cholesterol; GWAS, genome-wide association studies; LD, linkage disequilibrium; MR, Mendelian randomization; MVMR, multivariable MR.

**Figure 2 biomedicines-13-00723-f002:**
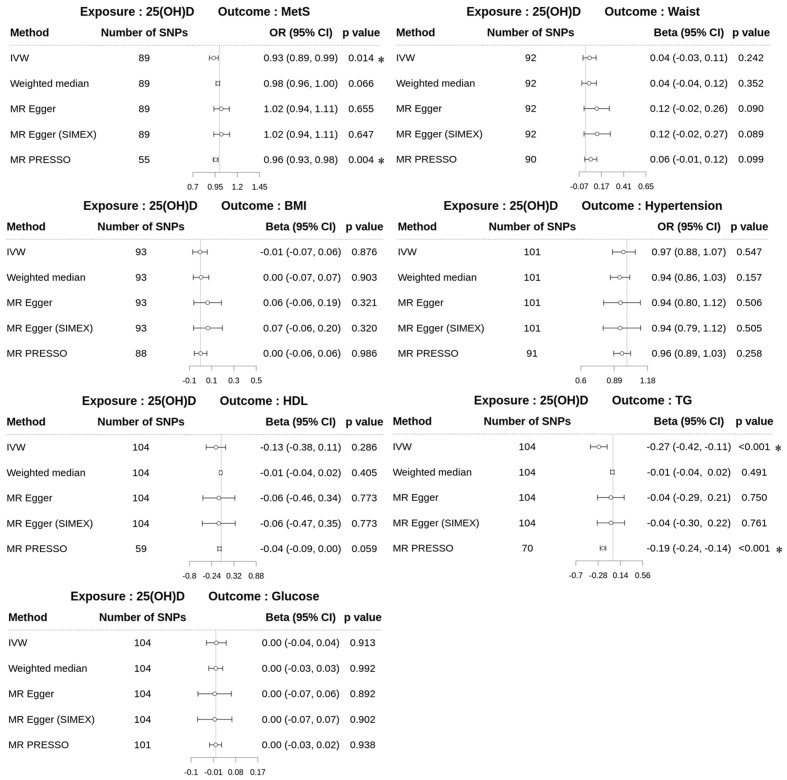
Forest plots of forward MR study (*dataset 2*) to explore the effect of 25(OH)D on MetS and MetS risk factors. 25(OH)D, 25-hydroxyvitamin D; MetS, metabolic syndrome; Waist, waist circumference; BMI, body mass index (weighted median; β 0.0044, MR–PRESSO; β 0.0005); HDL, high-density lipoprotein cholesterol; TG, triglyceride, glucose (IVW, β 0.0022; weighted median, β −0.0001, MR–Egger, β 0.0002; MR–Egger (SIMEX), β 0.0002; MR–PRESSO, β −0.0010); SNPs, single-nucleotide polymorphisms; IVW, inverse-variance weighted; MR, Mendelian randomization; PRESSO, pleiotropy residual sum and outlier; SIMEX, simulation extrapolation; OR, odds ratio; Beta, beta coefficient; CI, confidence interval; *, *p* < 0.05.

**Figure 3 biomedicines-13-00723-f003:**
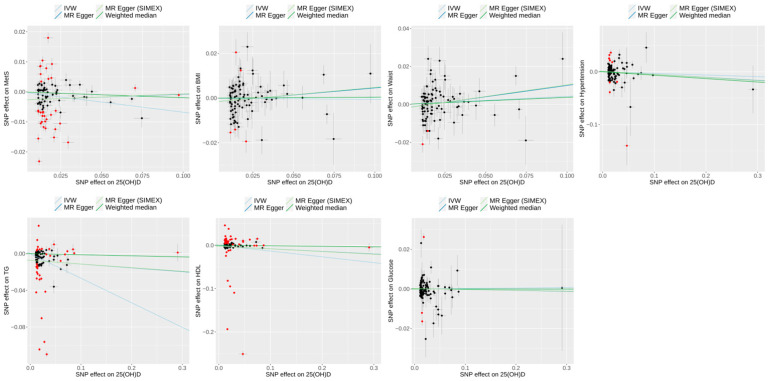
Scatter plots of forward MR study (*dataset 2*) exploring the effect of 25(OH)D on MetS and MetS risk traits. Light blue, dark blue, light green, and dark green regression lines represent the estimates from the IVW, MR–Egger, MR–Egger (SIMEX), and weighted median methods, respectively. Black dots were shown on SNPs as IVs. Red dots highlight the outliers identified in the MR–PRESSO analysis. 25(OH)D, 25-hydroxyvitamin D; MetS, metabolic syndrome; BMI, body mass index; Waist, waist circumference; TG, triglyceride; HDL, high-density lipoprotein cholesterol; SNPs, single-nucleotide polymorphisms; IVW, inverse-variance weighted; MR, Mendelian randomization; PRESSO, pleiotropy residual sum and outlier; SIMEX, simulation extrapolation.

**Figure 4 biomedicines-13-00723-f004:**
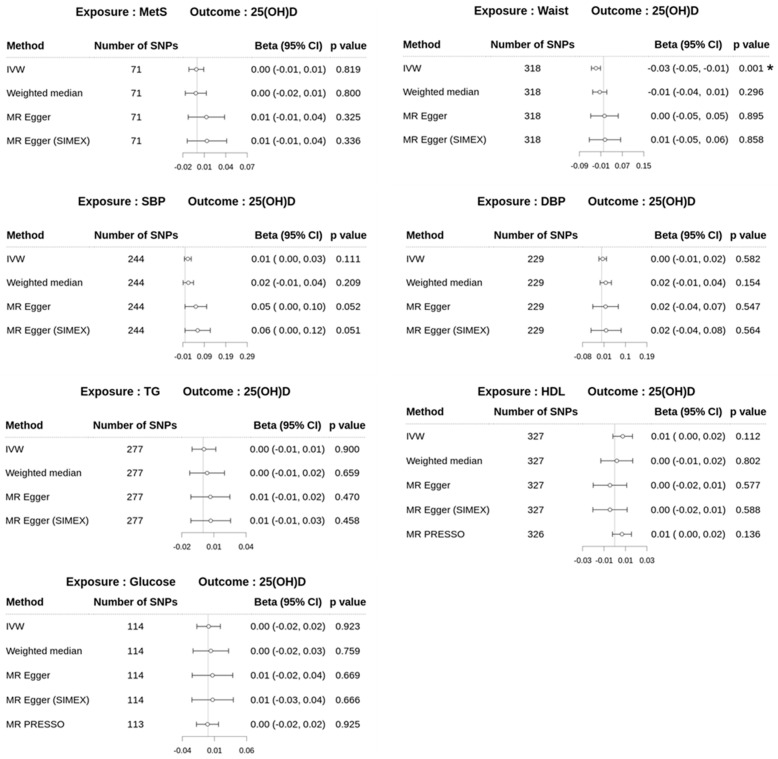
Forest plots of reverse MR study from *dataset 1* displaying the effect of MetS and MetS risk factors on 25(OH)D and risk factors. 25(OH)D, 25-hydroxyvitamin D; MetS, metabolic syndrome (IVW, β = −0.001; weighted median, β = −0.002; MR Egger, β = 0.003); Waist, waist circumference (MR–Egger, β = −0.0053); TG, triglyceride (IVW, β = 0.001; weighted median, β = 0.004); HDL, high-density lipoprotein cholesterol (weighed median, β = 0.002; MR Egger, β = −0.004; MR Egger (SIMEX), β = −0.004); SBP, systolic blood pressure; DBP, diastolic blood pressure (IVW, β = 0.004); glucose (IVW, β = 0.001; weighed median, β = 0.004; MR PRESSO, β = −0.001); SNPs, single-nucleotide polymorphisms; IVW, inverse-variance weighted; MR, Mendelian randomization; PRESSO, pleiotropy residual sum and outlier; SIMEX, simulation extrapolation; Beta, beta coefficient; CI, confidence interval; *, *p* < 0.05 significance.

**Figure 5 biomedicines-13-00723-f005:**
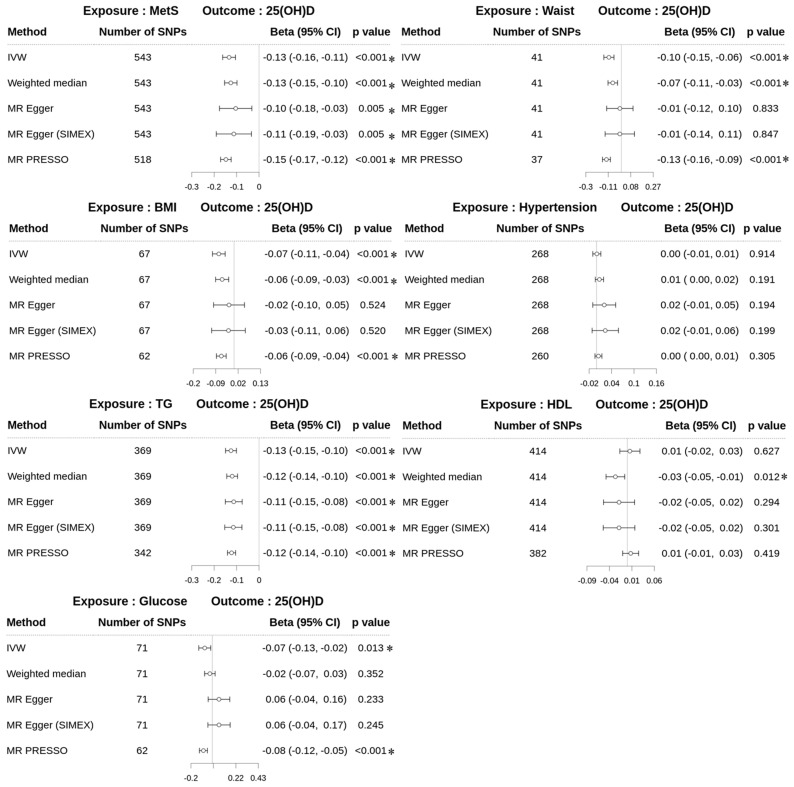
Forest plots of reverse MR study from *dataset 2* displaying the effect of MetS and MetS risk factors on 25(OH)D and risk factors. 25(OH)D, 25-hydroxyvitamin D; MetS, metabolic syndrome; Waist, waist circumference; BMI, body mass index; Hypertension (IVW, β = 0.001; MR PRESSO, β = 0.004); TG, triglyceride; HDL, high-density lipoprotein cholesterol; SNPs, single-nucleotide polymorphisms; IVW, inverse-variance weighted; MR, Mendelian randomization; PRESSO, pleiotropy residual sum and outlier; SIMEX, simulation extrapolation; Beta, beta coefficient; CI, confidence interval; *, *p* < 0.05.

**Figure 6 biomedicines-13-00723-f006:**
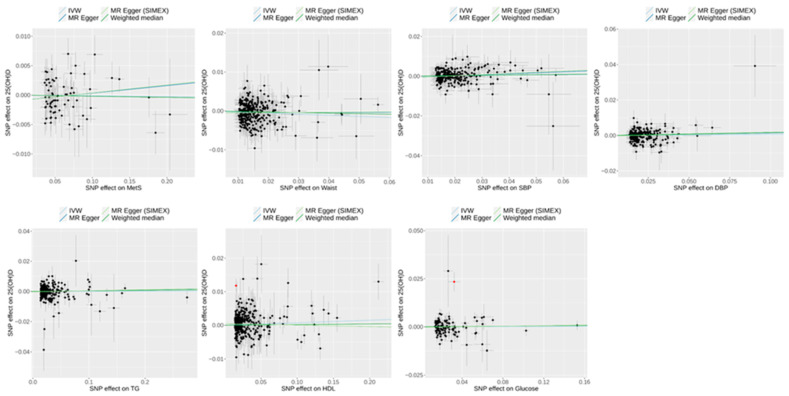
Scatter plots of reverse MR study (*dataset 1*) showing the effect of MetS and MetS risk traits on 25(OH)D. Light blue, dark blue, light green, and dark green regression lines represent the estimates from the IVW, MR–Egger, MR–Egger (SIMEX), and weighted median methods, respectively. Black dots were shown on SNPs as IVs. Red dots highlight the outliers identified in the MR-PRESSO analysis. 25(OH)D, 25-hydroxyvitamin D; MetS, metabolic syndrome; Waist, waist circumference; SBP, systolic blood pressure; DBP, diastolic blood pressure; TG, triglyceride; HDL, high-density lipoprotein cholesterol; SNPs, single-nucleotide polymorphisms; IVW, inverse-variance weighted; MR, Mendelian randomization; PRESSO, pleiotropy residual sum and outlier; SIMEX, simulation extrapolation.

**Figure 7 biomedicines-13-00723-f007:**
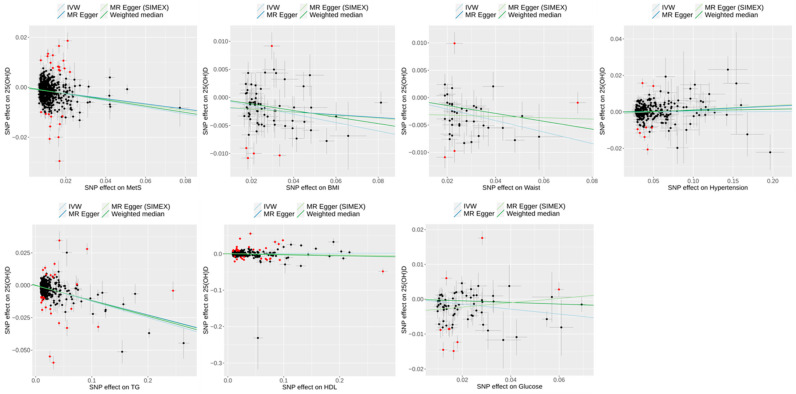
Scatter plots of reverse MR study (*dataset 2)* showing the effect of MetS and MetS risk traits on 25(OH)D. Black dots were shown on SNPs as IVs. Light blue, dark blue, light green, and dark green regression lines represent the estimates from the IVW, MR–Egger, MR–Egger (SIMEX), and weighted median methods, respectively. Red dots highlight the outliers identified in the MR-PRESSO analysis. 25(OH)D, 25-hydroxyvitamin D; MetS, metabolic syndrome; BMI, body mass index; Waist, waist circumference; TG, triglyceride; HDL, high-density lipoprotein cholesterol; SNPs, single-nucleotide polymorphisms; IVW, inverse-variance weighted; MR, Mendelian randomization; PRESSO, pleiotropy residual sum and outlier; SIMEX, simulation extrapolation.

**Table 1 biomedicines-13-00723-t001:** Summary statistics of data sources.

Traits	Data Sources	No. of Participants	Population	No. of Variants	Reference
** *Dataset 1* **					
25(OH)D level	SUNLIGHT Consortium	79,366	European	2,579,296	PMID: [40] 29343764
Metabolic syndrome	UK Biobank (UKB)	291,107	European	9,463,307	PMID: [39] 31589552
Waist circumference	UKB	419,807	European	23,861,814	ǂ
TG	UKB	400,639	European	23,861,718	
HDL	UKB	367,021	European	23,861,539	
SBP	UKB	396,663	European	23,861,710	
DBP	UKB	396,667	European	23,861,710	
Glucose	UKB	366,759	European	23,861,541	
** *Dataset 2* **					
25(OH)D level	UKB + European GWAS	443,734 (UKB: 401,460)	European	16,668,957	PMID: [41] 32059762
Metabolic syndrome	^(a)^ Multiple cohorts	1,384,348 *	European	2,265,555	PMID: [42] 39349817
Waist circumference	GIANT Consortium 2015	232,101	European	2,565,407	PMID: [43] 25673412
BMI	GIANT Consortium 2015	322,154	European	2,554,637	PMID: [44] 25673413
TG	GLGC Consortium	864,240	European	37,005,452	PMID: [45] 37237109
HDL	GLGC Consortium	888,227	European	36,588,494	PMID: [45] 37237109
Hypertension	FinnGen release 12	500,264	European	21,327,062	†
Glucose	MAGIC Consortium	200,622	European	34,064,006	PMID: [46] 34059833
** *Confounders, for reverse direction* **					
Skin color	UKB	415,018	European	9,463,307	ǂ
Physical activity (walking)	UKB	418,278	European	23,088,387	
** *Confounders, for forward direction* **					
Physical activity (strenuous sports)	UKB	418,278	European	22,111,708	ǂ
Alcohol consumption	UKB	419,936	European	21,143,063	
Smoking	UKB	418,817	European	22,122,417	

ǂ Data were obtained from https://pan.ukbb.broadinstitute.org/downloads/index.html (assessed 17 June 2024). † Data were obtained from https://finngen.gitbook.io/documentation/data-download (assessed 22 October 2024). * The effective sample size of the metabolic syndrome was estimated. ^(a)^ GIANT Consortium 2018 for BMI; MAGIC Consortium for Glucose; GLGC Consortium for HDL and TG; FinnGen release 7 and UKB for Hypertension; FinnGen release 7, meta-analysis of 32 cohorts, and Million Veteran Program for Type-2 diabetes; and UKB for Waist circumference. 25(OH)D, 25-hydroxyvitamin D; TG, triglyceride; HDL, high-density lipoprotein cholesterol; SBP, systolic blood pressure; DBP, diastolic blood pressure; BMI, body mass index.

**Table 2 biomedicines-13-00723-t002:** Heterogeneity and horizontal pleiotropy of instrumental variables in univariable MR.

Exposure	Outcome				Heterogeneity		Horizontal Pleiotropy				
					Cochran’s Q	Rücker’s Q’	MR-PRESSO Global Test	MR–Egger		MR–Egger (SIMEX)	
		N	F	I^2^ (%)	*p*-Value	*p*-Value	*p*-Value	Intercept, β (SE)	*p*-Value	Intercept, β (SE)	*p*-Value
** *Dataset 1 (reverse direction)* **											
Metabolic syndrome	25(OH)D level	71	65.21	89.27	0.044	0.049	0.042	−0.0010 (0.0008)	0.240	−0.0010 (0.0009)	0.252
WC		318	55.98	86.18	0.190	0.198	0.185	−0.0005 (0.0004)	0.198	−0.0006 (0.0005)	0.228
TG		277	128.43	98.11	<0.001	<0.001	<0.001	−0.0002 (0.0003)	0.415	−0.0003 (0.0003)	0.406
HDL		327	127.80	97.72	<0.001	<0.001	<0.001	0.0005 (0.0003)	0.067	0.0005 (0.0003)	0.072
SBP		244	57.37	90.79	0.070	0.079	0.072	−0.0008 (0.0005)	0.128	−0.0010 (0.0006)	0.111
DBP		229	56.92	82.45	0.039	0.036	0.040	−0.0003 (0.0006)	0.646	−0.0003 (0.0007)	0.647
Glucose		114	115.39	97.99	<0.001	<0.001	0.001	−0.0002 (0.0005)	0.647	−0.0002 (0.0005)	0.645
** *Dataset 2 (forward direction)* **											
25(OH)D level	Metabolic syndrome	89	114.98	97.72	<0.001	<0.001	<0.001	−0.0027 (0.0011)	0.012	−0.0027 (0.0011)	0.013
	WC	92	117.20	96.47	<0.001	<0.001	<0.001	−0.0020 (0.0015)	0.199	−0.0020 (0.0016)	0.192
	BMI	93	116.40	96.72	<0.001	<0.001	<0.001	−0.0018 (0.0014)	0.208	−0.0018 (0.0014)	0.208
	TG	104	118.64	97.46	<0.001	<0.001	<0.001	−0.0072 (0.0032)	0.028	−0.0072 (0.0033)	0.030
	HDL	104	118.64	97.46	<0.001	<0.001	<0.001	−0.0023 (0.0051)	0.650	−0.0023 (0.0052)	0.659
	HTN	101	124.84	97.79	<0.001	<0.001	<0.001	0.0008 (0.0022)	0.703	0.0009 (0.0022)	0.697
	Glucose	104	118.64	98.17	<0.001	<0.001	<0.001	0.0002 (0.0008)	0.805	0.0002 (0.0009)	0.813
** *Dataset 2 (reverse direction)* **											
Metabolic syndrome	25(OH)D level	543	74.23	88.79	<0.001	<0.001	<0.001	−0.0004 (0.0005)	0.385	−0.0003 (0.0005)	0.537
WC		41	59.80	87.11	<0.001	<0.001	<0.001	−0.0030 (0.0017)	0.087	−0.0029 (0.0018)	0.119
BMI		67	67.60	90.07	<0.001	<0.001	<0.001	−0.0016 (0.0011)	0.162	−0.0015 (0.0012)	0.211
TG		369	138.96	97.82	<0.001	<0.001	<0.001	−0.0004 (0.0005)	0.402	−0.0003 (0.0005)	0.446
HDL		414	147.02	98.17	<0.001	<0.001	<0.001	0.0008 (0.0004)	0.075	0.0008 (0.0004)	0.077
HTN		268	55.19	91.24	<0.001	<0.001	<0.001	−0.0009 (0.0007)	0.178	−0.0011 (0.0008)	0.181
Glucose		71	121.30	96.85	<0.001	<0.001	<0.001	−0.0035 (0.0012)	0.003	−0.0035 (0.0012)	0.004

Cochran’s Q test from IVW and Rücker’s Q’ test from MR–Egger test were performed for heterogeneity analysis. N, number of instruments; F, F statistic mean; IVW, inverse-variance weighted; MR, Mendelian randomization; PRESSO, pleiotropy residual sum and outlier; SIMEX, simulation extrapolation; β, beta coefficient; SE, standard error; 25(OH)D, 25-hydroxyvitamin D; WC, waist circumference; TG, triglyceride; HDL, high-density lipoprotein cholesterol; SBP, systolic blood pressure; DBP, diastolic blood pressure; BMI, body mass index.

**Table 3 biomedicines-13-00723-t003:** Multivariable MR IVW results.

Datasets	Exposure	Outcome	Model 1 β (95% CI)	Model 2 β (95% CI)
*Dataset 1* (*reverse direction*)	Metabolic syndrome	25(OH)D level	−0.002 (−0.018, 0.013)	−0.006 (−0.020, 0.008)
	Waist circumference		−0.032 (−0.070, 0.006)	−0.036 (−0.065, −0.006) *
	TG		−0.009 (−0.068, 0.050)	−0.010 (−0.068, 0.047)
	HDL		0.013 (−0.017, 0.042)	0.014 (−0.011, 0.040)
	SBP		−0.024 (−0.089, 0.041)	0.0001 (−0.026, 0.026)
	DBP		0.047 (−0.107, 0.201)	−0.015 (−0.040, 0.011)
	Glucose		−0.017 (−0.05, 0.015)	−0.022 (−0.057, 0.014)
*Dataset 2* (*forward direction*)	25(OH)D level	Metabolic syndrome	−0.030 (−0.145, 0.085)	−0.060 (−0.159, 0.040)
		Waist circumference	0.145 (−0.0004, 0.291)	0.106 (−0.028, 0.239)
		BMI	0.095 (−0.062, 0.251)	0.034 (−0.094, 0.163)
		TG	−0.118 (−0.388, 0.151)	−0.121 (−0.362, 0.120)
		HDL	−0.134 (−1.342, 1.074)	−0.111 (−0.329, 0.107)
		Hypertension	−0.107 (−0.286, 0.072)	−0.115 (−0.241, 0.011)
		Glucose	−0.010 (−0.034, 0.014)	−0.012 (−0.048, 0.024)
*Dataset 2* (*reverse direction*)	Metabolic syndrome	25(OH)D level	−0.135 (−0.175, −0.095) *	−0.141 (−0.175, −0.107) *
	Waist circumference		−0.087 (−0.135, −0.039) *	−0.097 (−0.143, −0.051) *
	BMI		−0.055 (−0.096, −0.015) *	−0.059 (−0.095, −0.024) *
	TG		−0.120 (−0.164, −0.076) *	−0.120 (−0.164, −0.077) *
	HDL		−0.024 (−0.084, 0.035)	−0.024 (−0.083, 0.035)
	Hypertension		0.006 (−0.015, 0.027)	0.007 (−0.014, 0.028)
	Glucose		−0.058 (−0.116, 0.001)	−0.058 (−0.118, 0.002)

Model 1 includes adjustments for all confounders: physical activity (strenuous sports), alcohol consumption, and smoking in the forward direction, and physical activity (walking) and skin color in the reverse direction. Model 2 includes only confounders with conditional F-statistics >5: smoking (forward direction) and skin color (reverse direction). * The 95% confidence interval does not include 0, signifying a statistically significant association. MR, Mendelian randomization; IVW, inverse-variance weighted; 25(OH)D, 25-hydroxyvitamin D; TG, triglyceride; HDL, high-density lipoprotein; SBP, systolic blood pressure; DBP, diastolic blood pressure; BMI, body mass index.

## Data Availability

The datasets for the GWAS summary statistics used in this analysis are available from the GWAS Catalog (https://www.ebi.ac.uk/gwas/summary-statistics, accessed on 23 October 2024), and Pan-UK Biobank (https://pan.ukbb.broadinstitute.org/downloads/index.html, accessed on 17 June 2024), FinnGen (https://finngen.gitbook.io/documentation/data-download, accessed on 22 October 2024), GIANT Consortium (https://portals.broadinstitute.org/collaboration/giant/index.php/GIANT_consortium_data_files, accessed on 29 November 2024), GLGC Consortium (https://csg.sph.umich.edu/willer/public/glgc-lipids2021/, accessed on 28 November 2024), and MAGIC Consortium (http://magicinvestigators.org/downloads/, accessed on 28 November 2024).

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
