# Peer review of "Causal Effects of 25-Hydroxyvitamin D on Metabolic Syndrome and Metabolic Risk Traits: A Bidirectional Two-Sample Mendelian Randomization Study"

_biomedicines, 2025, doi:10.3390/biomedicines13030723_

Round 1
Reviewer 1 Report (Previous Reviewer 1)
Comments and Suggestions for Authors
Review Paper: "Causal Effects of 25-hydroxyvitamin D on Metabolic Syndrome and Metabolic Risk Traits: A Bidirectional Two-Sample Mendelian Randomization Study"
Strengths:
Robust Mendelian Randomization (MR) Methodology:
This study used a comprehensive two-sample MR approach with validation through several statistical methods (IVW, MR-Egger, Weighted Median, MR-PRESSO).
The use of large datasets from GWAS and UK Biobank strengthened the results of the study.
Bidirectional Approach:
MR analysis was conducted in two directions to explore the possible causal relationship between 25(OH)D and metabolic syndrome and its risk factors.
This provides insight into the possibility that 25(OH)D levels are a consequence of metabolic syndrome, rather than a primary cause.
Multivariable Mendelian Randomization (MVMR):
By considering multiple covariates (physical activity, alcohol consumption, smoking), this study attempted to control for potential biases originating from environmental factors.
Data-Driven Conclusions:
The results do not suggest a direct causal relationship between 25(OH)D levels and metabolic syndrome, but rather an inverse relationship, with metabolic syndrome and its risk factors contributing to low 25(OH)D levels.
Weaknesses and Improvements:
More Context Needed on Biological Mechanisms
This study suggests that 25(OH)D levels may be a consequence of metabolic syndrome rather than a causal factor.
To strengthen this argument, more discussion is needed on the biological mechanisms underlying how obesity, insulin resistance, and other metabolic factors may lead to low vitamin D levels.
Studies such as Manoppo et al. (2022) in Frontiers in Nutrition have addressed the relationship of vitamin D to obesity and metabolic syndrome in the broader context of inflammation and lipid metabolism (DOI: 10.3389/fnut.2022.1025396). They should complement the discussion in this paper.
This study can be used as a reference to support the idea that the relationship between 25(OH)D and metabolic syndrome is more complex than a linear relationship.
Genetic Variability and Population Factors
This study was limited to a European population, so the results may not be fully generalizable to other populations with different patterns of vitamin D consumption and sun exposure.
It would be beneficial if this study addressed how these findings might differ in Asian or African populations, which have different genetic variations in vitamin D metabolism and environmental factors.
Discussion on Clinical Interventions
Since this study did not find a direct causal relationship between 25(OH)D and metabolic syndrome, there is an implication that vitamin D supplementation may not have significant benefits in preventing metabolic syndrome.
However, some studies suggest that vitamin D-based interventions may still be beneficial in certain groups, especially those with severe deficiency.
Related studies that address the benefits of vitamin D supplementation in populations with obesity and insulin resistance may also be considered.
Author Response
Comment 1: This study suggests that 25(OH)D levels may be a consequence of metabolic syndrome rather than a causal factor. To strengthen this argument, more discussion is needed on the biological mechanisms underlying how obesity, insulin resistance, and other metabolic factors may lead to low vitamin D levels. Studies such as Manoppo et al. (2022) in Frontiers in Nutrition have addressed the relationship of vitamin D to obesity and metabolic syndrome in the broader context of inflammation and lipid metabolism (DOI: 10.3389/fnut.2022.1025396). They should complement the discussion in this paper. This study can be used as a reference to support the idea that the relationship between 25(OH)D and metabolic syndrome is more complex than a linear relationship.
Response 1: Thank you for your insightful suggestion to strengthen the discussion of our manuscript by incorporating relevant insights from the opinion paper that you have suggested on the importance of biological mechanisms in the broader context of inflammation and lipid metabolism regarding 25(OH)D for metabolic diseases. We have revised and enhanced the discussion to address biological mechanisms in the broader context of inflammation and lipid metabolism regarding 25(OH)D for metabolic diseases. Key points from the reference article have been integrated to highlight the critical role of 25(OH)D in lipid metabolism and inflammation (page 15, Line 507-514), and cited it as the reference in the discussion, ensuring that readers have direct access to the source for more in-depth information.
“It is also important to consider the biological mechanisms that associate with MetS components such as obesity, insulin resistance, and inflammation to lower 25(OH)D levels, to strengthen our findings. Obesity has been known to lead 25(OH)D sequestration in adipose tissue to lower circulating levels of 25(OH)D [76]. Additionally, Manoppo et al. reported that the interplay among 25(OH)D, lipid metabolism, and inflammation may create a loop wherein metabolic dysfunction worsens 25(OH)D deficiency, suggesting that the relationship between 25(OH)D and MetS is more complex than a linear one [26].”
Comment 2: This study was limited to a European population, so the results may not be fully generalizable to other populations with different patterns of vitamin D consumption and sun exposure. It would be beneficial if this study addressed how these findings might differ in Asian or African populations, which have different genetic variations in vitamin D metabolism and environmental factors.
Response 2: We appreciate your opinion regarding the limit use of GWAS data from European population, rather than other populations. Genetic variations in 25(OH)D metabolism, dietary patterns, and differences in sun exposure significantly influence 25(OH)D levels across the populations. We have included in the discussion section how 25(OH)D levels and metabolic health differ across ethnic groups (page 15, Line 551-562).
“In this large-scale MR analysis, our results are restricted to a European population in order to minimize bias from a diverse, mixed population, which may limit the generalizability of our findings to other ethnic populations. Genetic variations in 25(OH)D metabolism, dietary patterns, and differences in sun exposure can significantly influence 25(OH)D status across populations [80-84]. For instance, in northern Chinese population, 25(OH)D levels were inversely related to the metabolic risk profiles, whereas in Asian Indians, who have a high prevalence of 25(OH)D deficiency, no significant association with metabolic syndrome, or insulin resistance was observed. Studies in East Africans address the complex interaction between 25(OH)D levels and metabolic health [85-87]. Future studies should aim to explore the relationship between 25(OH)D and MetS in more diverse populations.”
Comment 3: Since this study did not find a direct causal relationship between 25(OH)D and metabolic syndrome, there is an implication that vitamin D supplementation may not have significant benefits in preventing metabolic syndrome. However, some studies suggest that vitamin D-based interventions may still be beneficial in certain groups, especially those with severe deficiency. Related studies that address the benefits of vitamin D supplementation in populations with obesity and insulin resistance may also be considered.
Response 3: We appreciate your suggestion to provide more detailed descriptions for benefit from 25(OH)D supplementation. We have included references to recent clinical trials and meta-analyses that indicate potential improvements in insulin sensitivity and inflammation in population with obesity and diabetes in the discussion section (page 16, Line 562-565)
“Some studies suggest that clinical intervention with 25(OH)D may improve insulin sensitivity and reduce inflammation in individuals with obesity or type 2 diabetes [85,88]. This also emphasizes the need for further research to assess potential benefits in specific groups.”
Reviewer 2 Report (New Reviewer)
Comments and Suggestions for Authors
The paper "Causal Effects of 25-hydroxyvitamin D on Metabolic Syndrome and Metabolic Risk Traits: A Bidirectional Two-Sample Mendelian Randomization Study" is suitable for publication in the journal Biomedicines and within the scope of the journal.
It seems that the paper has been already revised and some major changes have been undertaken. In the present form it may be published with some minor corrections.
The Introduction is comprehensive and well written, it provides all the necessary information that are required to introduce the reader to the main issue of the study.
The methodology is precise and clear.
The results are clearly presented and both informative and visually attractive.
The conclusion is valid and correspodens to the aim of the study.
Minor comments:
Line 512: please delete the sentence „Our MR study had several strengths.“ You may start with something like: The main advantage of this study is... or In this study for the first time (and than explain the main points of your study) or even to start from the second point like :
In this two-sample MR study for the first time a large-scale genomic datasets from individuals of European ancestry was employed.
Just make a bold argument about this survey.
Line 521. please also delete: However, this study also had some limitations. Just start with the limitations.
Author Response
Comment 1: Line 512: please delete the sentence „Our MR study had several strengths. “ You may start with something like: The main advantage of this study is... or In this study for the first time (and then explain the main points of your study) or even to start from the second point like:
In this two-sample MR study for the first time a large-scale genomic datasets from individuals of European ancestry was employed.
Just make a bold argument about this survey.
Response 1: We appreciate this suggestion and have modified the sentence to make a bold argument about this survey (page 15, Line 551).
"The main advantage of this study is that it reinforces the presence of inverse causality of metabolic traits on 25(OH)D levels, providing plausible evidence from MVMR results using two different datasets (dataset 1 and dataset 2 for reverse direction). Additionally, to reduce the impact of population stratification, we utilized large-scale genomic datasets from individuals of European ancestry in a two-sample MR study. Furthermore, our study accounted for confounding factors by employing MVMR, which provides a framework to adjust for certain pleiotropic pathways and confounders, thereby enhancing the reliability of causal inference."
Comment 2: please also delete: However, this study also had some limitations. Just start with the limitations.
Response 2: We have revised the text accordingly by removing the phrase “However, this study also had some limitations.” and directly introducing the limitations.
This manuscript is a resubmission of an earlier submission. The following is a list of the peer review reports and author responses from that submission.
Round 1
Reviewer 1 Report
Comments and Suggestions for Authors
- Strengthen the introduction with this opinion paper and give urgency to the importance of vitamin D for metabolic diseases including fatty liver: https://doi.org/10.3389/fnut.2022.1025396
- Clearly outline the gap in research and how your study addresses it.
- Ensure each step is described in detail, and any protocols referenced are easily accessible.
Reviewer 2 Report
Comments and Suggestions for Authors
Dear editor and authors, after reviewing the article that has been sent to me, I consider it to be very suitable for publication. I believe that it is impeccably designed and executed. The objectives set and the results are of great interest. The presentation of the design and the results through tables and figures is exceptional. For all these reasons, I believe that it can be published in its current state.
Reviewer 3 Report
Comments and Suggestions for Authors
After careful review of the manuscript by Lee et al, which applies a two-sample Mendelian Randomization (MR) to evaluate potential causal link between 25-OH Vitamin D and Metabolic Syndrome (MetS), I cannot support its publication in the present form.
Mendelian Randomization studies can be powerful research tools to understand casualty but are susceptible to compromised validity when based on “weak instruments” (Skrivankova et al, 2021). Additionally, excluding confounding factors should be extensively validated when measuring outcomes that are largely affected by environmental factors, such as vitamin D levels (dependent on sun exposure and intake) and Metabolic Syndrome (varying largely by socioeconomic state and access to calories).
The presented study relies on genetic instruments (single nucleotide polymorphism -SNPs) associated with 25(OH) Vitamin D levels derived from the SUNLIGHT Consortium by Jiang et al (2018). However, Jiang et al (2018) estimated that “only 2.84% of the variation in blood levels of 25-hydroxyvitamin D could be explained by known genetic regions”, further highlighting that “consideration of genetic background is not required when determining population based vitamin D intake recommendations”, since genetics are proven to have minimal explanatory power. Contrary, it is acknowledged that environmental factors such as age, BMI, season of blood drawn, vitamin D dietary intake, vitamin D supplement intake, region of residence and ethnicity have a substantial influence, explaining ~18% of the observed variance (Hiraki et al 2013).
Even though robust statistical analysis were used in the next steps of the research, this root issue is considered a major limitation and was not effectively communicated, neither in methodology nor in discussion. The low variance explained by the genetic variants used, is not compensated with high F-values or detailed assessments for heterogeneity and horizontal pleiotropy.
In conclusion, this manuscript should be re-examined for publication only after important revision, including restructuring the conceptual design and methodological approach.
References
Hiraki LT, Major JM, Chen C, Cornelis MC, Hunter DJ, Rimm EB, Simon KC, Weinstein SJ, Purdue MP, Yu K, Albanes D, Kraft P. Exploring the genetic architecture of circulating 25-hydroxyvitamin D. Genet Epidemiol. 2013 Jan;37(1):92-8. doi: 10.1002/gepi.21694. Epub 2012 Nov 7. PMID: 23135809; PMCID: PMC3524394.
Jiang X, O'Reilly PF, Aschard H, Hsu YH, Richards JB, Dupuis J, Ingelsson E, Karasik D, Pilz S, Berry D, et al. Genome-wide association study in 79,366 European-ancestry individuals informs the genetic architecture of 25-hydroxyvitamin D levels. Nat Commun. 2018 Jan 17;9(1):260. doi: 10.1038/s41467-017-02662-2. PMID: 29343764; PMCID: PMC5772647.
Skrivankova VW, Richmond RC, Woolf BAR, Davies NM, Swanson SA, VanderWeele TJ, Timpson NJ, Higgins JPT, Dimou N, Langenberg C, et al. Strengthening the reporting of observational studies in epidemiology using mendelian randomisation (STROBE-MR): explanation and elaboration. BMJ. 2021 Oct 26;375:n2233. doi: 10.1136/bmj.n2233. PMID: 34702754; PMCID: PMC8546498.